

# Synthesis and biological evaluation of small molecule modulators of CDK8/Cyclin C complex with phenylaminoquinoline scaffold

Mohammad M. Al-Sanea

Pharmaceutical Chemistry Department, College of Pharmacy, Jouf University, Sakaka, Aljouf, Saudi Arabia

## ABSTRACT

**Background.** CDK8/CycC complex has kinase activity towards the carboxyterminal domain of RNA polymerase II, and contributes to the regulation of transcription via association with the mediator complex. Different human malignancies, mainly colorectal and gastric cancers, were produced as a result of overexpression of CDK8/CycC in the mediator complex. Therefore, CDK8/CycC complex represents as a cancer oncogene and it has become a potential target for developing CDK8/CycC modulators.

**Methods.** A series of nine 4-phenylaminoquinoline scaffold-based compounds **5a-i** was synthesized, and biologically evaluated as potential CDK8/CycC complex inhibitors.

**Results.** The scaffold substituent effects on the intrinsic inhibitory activity toward CDK8/CycC complex are addressed trying to present a novel outlook of CDK8/CycC Complex inhibitors with 4-phenylaminoquinoline scaffold in cancer therapy. The secondary benzenesulfonamide analogues proved to be the most potent compounds in suppressing CDK8/CycC enzyme, whereas, their primary benzenesulfonamide analogues showed inferior activity. Moreover, the benzene reversed sulfonamide analogues were totally inactive.

**Discussion.** The titled scaffold showed promising inhibitory activity data and there is a crucial role of un/substituted sulfonamido group for CDK8/CycC complex inhibitory activity. Compound **5d** showed submicromolar potency against CDK8/CycC ($IC_{50}$ = 0.639 $\mu$M) and it can be used for further investigations and to design another larger library of phenylaminoquinoline scaffold-based analogues in order to establish detailed SARs.

## INTRODUCTION

Cyclin-dependent kinases (CDKs) drive cell cycle through phosphorylation of a variety of vital substrates (*Satyanarayana & Kaldis, 2009*). Association of CDKs with regulatory partners (cyclins) regulates CDKs activity (*Obaya & Sedivy, 2002*). Therefore, several cyclin/kinase complexes have been considered as essential for controlled cell proliferation (*Sears & Nevins, 2002*). Cyclin C is known to form a stable complex with CDK8 (CDK8/CycC complex). The kinase active complex is associated with direct

Corresponding author
Mohammad M. Al-Sanea,
mmalsanea@ju.edu.sa

phosphorylation activity towards gene-specific transcription factors, thus controls their downstream function (*Nemet et al., 2014*). Hence, CDK8/CycC can modulate transcriptional output from distinct transcription factors involved in oncogenic control (*Malik & Roeder, 2005*). Recent evidence supports the idea that mediator complex-associated CDK8/CycC has been involved in the regulation of multiple transcription pathways and implicated as an oncogene in colorectal and gastric cancers through activation of WNT signaling (*Kim et al., 2006*; *Rzymski et al., 2015*). CDK8 is amplified and overexpressed in colon, gastric, breast cancers and melanoma (*Li et al., 2014a*; *Roninson et al., 2019*). Accordingly, CDK8/CycC complex may represent a potential drug target for different kinds of human malignancies with reduced toxic effects on normal cells (*Chen, Ren & Chang, 2019*; *He et al., 2019*; *Rzymski et al., 2015*; *Sánchez-Martínez et al., 2019*; *Schneider et al., 2013*). Moreover, CDK8/CycC complex plays several roles in modulating gene expression levels (*Firestein et al., 2008*; *Knuesel et al., 2009a*; *Knuesel et al., 2009b*; *Li et al., 2014b*).

Even though CDK inhibitors have been abundantly described, attempts of discovering selective CDK8 inhibitors have emerged as a promising strategy for cancer therapy as Pan-CDK inhibitor has shown narrow therapeutic window and potential risks (*Al-Sanea et al., 2015a*; *Al-Sanea et al., 2015b*; *Al-Sanea et al., 2016b*; *Firestein et al., 2008*; *Kapoor et al., 2010*; *Xu & Ji, 2011*). Such selective inhibitors allow cancer therapy by reducing mitogenic signals in several cancer cells (*Adler et al., 2012*; *McDermott et al., 2017*; *Rzymski et al., 2015*).

Several chemical scaffold-based small molecules have been applied for the design of CDK8 inhibitors. Among these scaffolds, quinoline and its bioisosteres have successively shown potent modulation of CDK8 activity. The steroidal natural product cortistatin A, which has quinoline moiety as a hinge component and steroidal core responsible for extensive intermolecular forces with the ATP-binding cavity, showed a highly potent ATP-competitive CDK8 inhibitory activity ($IC_{50}$ = 15 nM) that exhibited anticancer activity in animal models of acute myeloid leukemia (AML) (*Cee et al., 2009*; *Crown, 2017*; *Pelish et al., 2015*; *Rzymski et al., 2017*). Senexin B with 4-aminoquinazoline scaffold showed potent CDK8 modulation with an $IC_{50}$ value of 24 nM (*McDermott et al., 2017*; *Rzymski et al., 2015*). In 2016, *Schiemann et al. (2016)* described new potent and selective CDK8 ligands with benzylindazole scaffold that showed an $IC_{50}$ value against CDK8 of 10 nM. Besides, many well-known kinase ligands such as sorafenib and imatinib represent another type of potent CDK8 inhibitors with different binding modes (*Chen, Ren & Chang, 2019*; *Schneider et al., 2011*).

It is noteworthy that the kinase activity of CDK8 is affected by substrate binding and association with other mediator complex members as well. By utilizing a scaffold hopping strategy on the aforementioned quinoline isosteres, a series of new phenylaminoquinoline derivatives with sulphonamide moiety at position 3 in terminal phenyl ring was designed, synthesized and pharmacologically evaluated as potential small molecule modulators of CDK8/CycC complex.

## MATERIALS & METHODS

All chemical reagents and solvents were of analytical grade, purchased from commercial suppliers (Sigma-Aldrich and Alfa-aesar) and utilized to accomplish this work as received. All reactions were carried out in a dry nitrogen atmosphere. The microwave-assisted synthesis was carried out in a Biotage Initiator apparatus operating in single mode, the microwave cavity producing controlled irradiation at 2.45 GHz (Biotage AB, Uppsala, Sweden). The reactions were run in sealed vessels. These experimentations were carried out by employing magnetic stirring and a fixed hold time applying variable power to reach (during 1 −2 min) and then keep the desired temperature in the vessel for the preset time. On the reactor vial glass, an IR sensor was applied to monitor the temperature. The NMR spectra were obtained on Bruker Avance 400 (400 MHz $^1$H and 100.6 MHz $^{13}$C NMR). Column chromatography was performed on Merck Silica Gel 60 (230–400 mesh). Thin layer chromatography (TLC) was performed using sheets pre-coated with silica gel 60 F254 supplied from Merck. The purity of compounds was determined by analytical high performance liquid chromatography (HPLC) using a Water ACQUITY UPLC (CORTECSTM) with C18 column (2.1 mm ×100 mm; 1.6 µm) at 40 °C. HPLC data were noted using parameters as follows: 0.1% formic acid in water and 0.1% formic acid in methanol and flow rate of 0.3 mL/min. Waters ACQUITY UPLC BEH C18 1.7 µ–Q-TOF SYNAPT G2-Si High Definition Mass Spectrometry was used to obtain high-resolution spectra. Compounds **3–4** and **5a–c, i** were previously reported (*Al-Sanea et al., 2019*).

Common procedures for the synthesis of key intermediates **4a–d.**

A solution of intermediated **3a–d** (1.0 mmol) in $POCl_3$ (six mL) was refluxed for 1 h. Evaporation of the mixture was performed in *vacuo* and the residue was extracted with methylene chloride, aqueous ammonia, and crushed ice. The methylene chloride layer was dried over anhydrous $Na_2SO_4$ and concentrated. Column chromatography ($SiO_2$, EA: *n*-Hex) was applied to purify the residue to get compounds **4a-c** (Scheme 1).

Common procedures for the synthesis of the target compounds **5a-i**

To a microwave vial, were sequentially added the appropriate intermediated **4a-c** (0.21 mmol), 3-amino-*N*-methylbenzenesulfonamide (0.04 gm, 0.21 mmol), or *N*-(3-phenyl)methanesulfonamide (0.04 gm, 0.21 mmol), 3-aminobenzenesulfonamide (0.036 gm, 0.21 mmol), and absolute ethyl alcohol (12 mL). The microwave vial was sealed and heated under microwave conditions at 150 °C for 30 min. The reaction mixture was evaporated *in vacuo* and the residue was extracted with ethyl acetate and $NaHCO_3$ (aq). The ethyl acetate layer was dried over anhydrous $Na_2SO_4$ and concentrated. The residue was purified by column chromatography ($SiO_2$, EA: *n*-Hex) to provide quinolines **5a-i** (Scheme 1).

Ethyl 6-Bromo-4-(3-sulfamoyl-phenylamino)-quinoline-3-carboxylate (**5a**)

Yellow solid, yield: 71%, mp: 235.6 −237.2 °C; $^1$H NMR (DMSO-*d6*, 400 MHz) $\delta$ *ppm*: 1.07 (t, 3H, $J = 5.6$ Hz, $CH_2$ $CH_3$), 3.88 (q, 2H, $J = 5.6$ Hz, $CH_2CH_3$), 7.17 (s, 1H), 7.36 (s, 2H, $SO_2NH_2$), 7.45–7.51 (m, 3H), 7.94 (s, 2H), 8.53 (s, 1H), 8.92 (s, 1H), 9.71 (s, 1H, NH); $^{13}$C NMR (DMSO-*d6*, 100 MHz) $\delta$ *ppm*: 14.22 ($CH_3$), 61.49 ($CH_2$), 111.61 (phenyl C-2),

**Scheme 1** **Synthesis of target quinolines 5a-i.** Reagents and conditions: (i) Diethyl ethoxymethylen-emalonate/Ethanol/reflux 1 h; (ii) Diphenyl ether/250 °C/45 min; (iii) POCl₃/reflux 1 h; (iv) Absolute ethanol/ reflux 4 h.

116.40 (phenyl C-4), 119.81 (phenyl C-6), 119.95 (quinoline C-6), 121.47 (quinoline C-3), 123.08 (quinoline C-10), 126.60 (quinoline C-5), 130.21 (phenyl C-5), 132.13 (quinoline C-8), 134.76 (quinoline C-7), 144.01 (phenyl C-1), 145.68 (phenyl C-3), 146.37 (quinoline C-2), 148.91(quinoline C-9), 152.06 (quinoline C-4), 166.33 (C =O); HRESI-MS $m/z$ calcd for $[M+H]^+$ $C_{18}H_{16}BrN_3O_4S$: 450.0123, found: 450.0127.

**Ethyl 6-methoxy-4-((3-sulfamoylphenyl)amino)quinoline-3-carboxylate (5b)**

Yellow solid, yield: 65%, ¹H NMR (DMSO-$d6$, 400 MHz at 298 K): 1.13 (t, 3H, $J = 6.8$ Hz, $CH_2CH_3$), 3.73 (s, 3H, $OCH_3$), 3.99 (q, 2H, $J = 6.8$ Hz, $CH_2CH_3$), 7.14–7.16 (m, 1H, phenyl H-2), 7.33 (s, 2H, phenyl H-4), 7.42–7.48 (m, 5H, phenyl H-5,6, quinoline H-5,7,8), 7.91–7.93 (m, 1H, quinoline H-2), 8.84 (s, 1H, NH); ¹³C NMR (DMSO-$d6$): 14.31 ($CH_2$ $CH_3$), 55.92 ($OCH_3$), 61.40 ($CH_2$), 103.51 (quinoline C-5), 111.54 (phenyl C-2), 116.36 (phenyl C-4), 119.51 (phenyl C-6), 121.78 (quinoline C-3), 122.35 (quinoline C-10), 123.78 (quinoline C-7), 130.17 (phenyl C-5), 131.56 (quinoline C-8), 144.36 (quinoline C-9), 145.55 (phenyl C-3), 146.16 (phenyl C-1), 146.59 (quinoline C-2), 148.90 (quinoline C-3), 157.51 (quinoline C-6), 166.90 (C =O); HRESI-MS calcd for $[M+H]^+$ $C_{19}H_{19}N_3O_5S$: 402.1124, found: 402.1116.

**Ethyl 7-chloro-6-fluoro-4-((3-sulfamoylphenyl)amino)quinoline-3-carboxylate (5c)**

Yellowish white solid, yield: 66%, ¹H NMR (DMSO-$d6$, 400 MHz at 298 K): 1.08 (t, 3H, $J = 7.2$ Hz, $CH_2CH_3$), 3.91 (q, 2H, $J = 7.2$ Hz, $CH_2CH_3$), 7.18–7.20 (m, 1H, quinoline H-2), 7.45–7.49 (m, 3H, phenyl H-4,5,6), 7.36 ( s, 2H), 8.18 (d, 1H, $J = 11.2$ Hz, quinoline H-8), 8.25 (d, 1H $J = 7.6$ Hz, quinoline H-7), 8.91 ( s, 1H, $SO_2NH$), 9.68 (s, 1H, NH); ¹³C NMR (DMSO-$d6$): 14.22 ($CH_2$ $CH_3$), 61.58 ($CH_2$), 110.16, 110.40 (phenyl C-2), 111.47, 116.52 (phenyl C-4), 120.18, 121.24 (phenyl C-6), 121.85 (quinoline

C-10), 124.41 (quinoline C-8), 130.29 (phenyl C-5), 131.58, 143.76 (phenyl C-3), 145.69 (phenyl C-1), 147.05 (quinoline C-2), 147.52 (quinoline C-5), 152.27 (quinoline C-3), 153.76 (quinoline C-7), 156.21 (quinoline C-6), 166.26 (C $=$O); HRESI-MS m/z calcd for [M+H]+ $C_{18}H_{15}ClFN_3O_4S$: 424.0534, found: 424.0525.

Ethyl 6-Bromo-4-((3-(N-methylsulfamoyl)phenyl)amino)quinoline-3-carboxylate (**5d**)

White solid, yield: 64%, mp: 212.9 −214.3 °C, [1]H NMR (DMSO-$d6$, 400 MHz) $\delta$ $ppm$: 1.08 (t, 3H, $J = 6.8$ Hz, $CH_2CH_3$), 2.98 (s, 3H, $NHCH_3$) 3.91 (q, 2H, $J = 6.8$ Hz, $CH_2CH_3$), 7.27 (d, 1H, $J = 7.6$ Hz, benzenesulfonamide H-2), 7.95 (m, 1H, quinoline H-5), 8.40 (s, 1H, quinoline H-8), 8.48 (s, 1H, quinoline H-2), 8.89 (s, 1H, $SO_2NH$), 9.64 (s, 1H, NH); [13]C NMR (DMSO-$d6$, 100 MHz) $\delta$ $ppm$: 14.20 ($CH_3$), 29.05 (N-$CH_3$), 61.45 ($CH_2$), 111.61 (benzenesulfonamide C-2), 117.40 (benzenesulfonamide C-4), 119.76 (benzenesulfonamide C-6), 121.02 (quinoline C-3), 122.53 (quinoline C-10), 123.04 (quinoline C-6), 126.69 (quinoline C-5), 130.56 (benzenesulfonamide C-5), 132.14 (quinoline C-8), 134.75 (quinoline C-7), 140.80 (benzenesulfonamide C-3), 144.35 (benzenesulfonamide C-1), 146.54 (quinoline C-2), 148.95 (quinoline C-9), 152.03 (quinoline C-4), 166.34 (C $=$O); HRESI-MS $m/z$ calcd for [M+H]$^+$ $C_{19}H_{19}BrN_3O_4S$: 464.0280, found: 464.0273.

Ethyl 6-methoxy-4-((3-(methylsulfonamido)phenyl)amino)quinoline-3-carboxylate (**5e**)

Yellow solid, yield: 64%, [1]H NMR (DMSO-$d6$, 400 MHz at 298 K): 1.17 (t, 3H, $J = 7.2$ Hz, $CH_2CH_3$), 2.96 (s, 3H, $SO_2CH_3$), 3.71 (s, 3H, $OCH_3$), 3.99 (q, 2H, $J = 7.2$ Hz, $CH_2CH_3$), 6.75–6.77 (m, 1H, phenyl H-2), 6.88–6.90 (m, 2H, phenyl H-4,6), 7.22–7.26 (m, 1H, phenyl H-5), 7.40–7.45 (m, 2H, quinoline H-5,7), 7.89 (d, $J = 8$ Hz, 1H, quinoline H-8), 8.80 (s, 1H, quinoline H-2), 9.52 ( s, 1H, $SO_2NH$), 9.74 (s, 1H, NH); [13]C NMR (DMSO-$d6$): 14.35 ($CH_2CH_3$), 55.82 ($OCH_3$) 61.29 ($CH_2$), 103.88 (quinoline C-5), 110.41 (phenyl C-2), 111.21 (phenyl C-4), 114.27 (phenyl C-6),115.58 (quinoline C-3),121.85 (quinoline C-10), 123.45 (quinoline C-7), 130.44 (phenyl C-5), 139.84 (quinoline C-8), 144.51 (quinoline C-9), 146.11 (phenyl C-3), 147.58 (phenyl C-1), 148.88 (quinoline C-2), 157.14 (quinoline C-3), 150.45 (quinoline C-6), 167.41(C $=$ O); HRESI-MS m/z calcd for [M+H]$^+$ $C_{20}H_{21}N_3O_5S$: 416.1280, found: 416.1278.

Ethyl 7-chloro-6-fluoro-4-((3-(methylsulfonamido)phenyl)amino)quinoline-3-carboxylate (**5f**)

Yellow solid, yield: 55%, [1]H NMR (DMSO-$d6$, 400 MHz at 298 K): 1.12 (t, 3H, $J = 6.8$ Hz, $CH_2CH_3$), 2.99 (s, 3H, $SO_2CH_3$), 3.92 (q, 2H, $J = 6.8$ Hz, $CH_2CH_3$), 6.77 (d, 1H, $J = 8$ Hz, phenyl H-2), 6.90–6.93 (m, 2H, phenyl H-4,6), 7.23–7.27 (m, H, phenyl H-5), 8.12 (d, 1H, $J = 11.2$ Hz, quinoline H-7), 8.21 (d, $J = 6$ Hz, 1H, quinoline H-8), 8.87 (s, 1H, quinoline H-2), 9.63 (s, 1H, $SO_2NH$), 9.79 (s, 1H, NH); HRESI-MS calcd for [M+H]$^+$ $C_{19}H_{17}ClFN_3O_4S$: 438.0691, found: 438.0687.

Ethyl 6-Bromo-4-(3-methanesulfonylphenylamino)-quinoline-3-carboxylate (**5g**)

Yellow solid, yield: 67%, mp: 183.4 −184.5 °C, [1]H NMR (DMSO-$d6$, 400 MHz) $\delta$ $ppm$: 1.12 (t, 3H, $J = 7.2$ Hz, $CH_2CH_3$), 2.98 (s, 3H, $SO_2CH_3$) 3.93 (q, 2H, $J = 7.2$ Hz, $CH_2CH_3$), 6.78 (s, 1H, phenyl H-2), 6.91–6.94 (m, 2H, phenyl H-4,6), 7.24 (m, 1H, phenyl H-5), 7.90

(m, 2H, quinoline H-5,7), 8.40 (m, 1H, quinoline H-8), 8.88 (s, 1H, quinoline H-2), 9.67 (s, 1H, NH); $^{13}$C NMR (DMSO-$d6$, 100 MHz) $\delta$ $ppm$: 14.29 (CH$_3$), 39.35 (SO$_2$CH$_3$), 61.41 (CH$_2$), 110.54, (phenyl C-2), 111.07 (phenyl C-4), 114.67 (phenyl C-6), 115.17 (quinoline C-7), 119.24 (quinoline C-3), 122.54 (quinoline C-10), 127.01 (quinoline C-6), 130.54 (quinoline C-5), 132.07 (phenyl C-5), 134.59 (quinoline C-8), 140.10 (phenyl C-3), 144.10 (phenyl C-1), 147.47 (quinoline C-2), 148.96 (quinoline C-9), 151.91 (quinoline C-4), 166.90 (C =O); HRESI-MS $m/z$ calcd for [M+H]$^+$ C$_{19}$H$_{19}$BrN$_3$O$_4$S: 464.0280, found: 464.0276.

Ethyl 6-methoxy-4-((3-(methylsulfonyl)phenyl)amino)quinoline-3-carboxylate (**5h**)

Yellow solid, yield: 62%,$^1$H NMR (DMSO-$d6$, 400 MHz at 298 K): 1.13 (t, 3H, $J$ = 7.2 Hz, CH$_2$CH$_3$), 2.39 (s, 3H, NHCH$_3$), 3.75 (s, 3H, OCH$_3$), 3.99 (q, 2H, $J$ = 7.2 Hz, CH$_2$CH$_3$), 7.24 (d, 1H, $J$ = 7.6 Hz, phenyl H-2), 7.36–7.52 (m, 6H, phenyl H-4,5,6, quinoline H-5,7,8), 7.95 (s, 1H, quinoline H-2), 8.85 (s, 1H, SO$_2$NH), 9.57 (s, 1H, NH); $^{13}$C NMR (DMSO-$d6$): 14.28 (CH$_2$ CH$_3$), 28.99 (NHCH$_3$), 55.91 (OCH$_3$) 61.35 (CH$_2$), 103.47 (quinoline C-5), 111.69 (phenyl C-2), 117.23 (phenyl C-4), 120.47 (phenyl C-6), 122.39 (quinoline C-3), 122.52 (quinoline C-10), 123.73 (quinoline C-7), 130.49 (phenyl C-5), 131.60 (quinoline C-8), 140.72 (quinoline C-9), 144.77 (phenyl C-3), 146.17 (phenyl C-1), 146.50 (quinoline C-2), 148.92 (quinoline C-3), 157.54 (quinoline C-6), 166.86 (C =O); HRESI-MS m/z calcd for [M+H]$^+$ C$_{20}$H$_{21}$N$_3$O$_5$S: 416.1280, found: 416.1277.

Ethyl 7-chloro-6-fluoro-4-((3-(methylsulfonyl)phenyl)amino)quinoline-3-carboxylate (**5i**)

Yellow solid, yield: 58%, $^1$H NMR (DMSO-$d6$, 400 MHz at 298 K): 1.07 (t, 3H $J$ = 6.8 Hz, CH$_2$CH$_3$), 2.40 (s, 3H, NHCH$_3$), 3.90 (q, 2H, $J$ = 6.8 Hz, CH$_2$CH$_3$), 7.27 (d, 1H, $J$ = 7.6 Hz phenyl H-2), 7.41(s, 1H, phenyl H-5), 7.49–7.53 (m, 2H, phenyl H-4,6), 8.19 (d, 1H, $J$ = 11.6 Hz quinoline H-7), 8.26 (d, 1H, $J$ = 7.2 Hz quinoline H-8), 8.91 (s, 1H, quinoline H-2), 9.70 (s, 1H, SO$_2$NH), 9.79 (s, 1H, NH); $^{13}$C NMR (DMSO-$d6$): 14.19 (CH$_2$ CH$_3$), 29.00 (NHCH$_3$), 61.52 (CH$_2$), 110.18, 110.42 (phenyl C-2), 111.56,117.44 (phenyl C-4), 121.17, 121.25 (phenyl C-6), 121.33 (quinoline C-10), 122.63 (quinoline C-8), 125.22, 125.43,130.61 (phenyl C-5), 131.58, 140.89, 144.09 (phenyl C-3), 146.97 (phenyl C-1), 147.02 (quinoline C-2), 147.54 (quinoline C-5), 152.26 (quinoline C-3), 153.76 (quinoline C-7), 156.21 (quinoline C-6), 166.24 (C = O). HRESI-MS calcd for [M+H]$^+$ C$_{19}$H$_{17}$Cl FN$_3$O$_4$S: 438.0691, found: 438.0693.

## In vitro kinase inhibition assay

Reaction Biology Corp. Kinase HotSpotSM service (http://www.reactionbiology.com) was used for the screening of tested compounds. Kinase Profiling is 10 dose IC$_{50}$ singlet assay. Activity of kinases were assessed by the HotSpot assay platform, which contained specific kinase/substrate pairs along with required cofactors (*Abdelazem et al., 2015*; *Abdelazem et al., 2016*; *Al-Sanea et al., 2016a*; *Al-Sanea et al., 2015a*; *Al-Sanea et al., 2015b*; *Al-Sanea, El-Deeb & Lee, 2013*; *Park et al., 2014*).

## Docking studies

Molecular Operating Environment (MOE version 2008.10) by Chemical Computing Group (CCG) was used for the docking studies (*Inc. CCG, 2016*). The protein preparation steps involved 3D protonation, energy minimization, and active site identification. The X-ray crystallographic structure of CDK8/CycC enzyme co-crystallized with Senexin A (PDB code 4f7s) was obtained from the Protein Data Bank (*Schneider et al., 2013*). The enzyme was prepared for virtual docking studies where: (i) the ligand molecule with any existing solvent molecules were removed. (ii) Hydrogen atoms were added to the structure with their standard geometry. To visualize the binding pocket, alpha spheres were created followed by the generation of dummy atoms on the centers of these spheres. The pocket was found to be a deep cavity lined with the amino acid residues including both hydrophobic and hydrophilic amino acids. Energy minimization tool MOPAC 7.0 was applied for the tilted compounds. (iii) MOE Alpha Site Finder was used for the active sites search and dummy atoms were created from the obtained alpha spheres. The obtained ligand-enzyme complex model was then used in calculating the energy parameters using MMFF94x force field energy calculation and predicting the ligand-enzyme interactions.

# RESULTS

## Chemistry

The methods followed for the synthesis of the target compounds **5a-i** are represented in Scheme 1. Anilines **1a–c** were firstly refluxed in ethanol with diethyl ethoxymethylen-emalonate to provide substituted phenylaminomethylenemalonates **2a–c.** Compounds **2a-c** were cyclized thermally in diphenyl ether to the corresponding 4-oxo-1,4-dihydroquinolines **3a–c**. Under the anhydrous condition, quinolines **3a–c** were chlorinated via heating with excess of $POCl_3$ to provide the key intermediates **4a–c**, as reported previously (*Al-Sanea et al., 2019*; *Medapi et al., 2015*; *Rivilli et al., 2018*). The target compounds **5a–i** were achieved through microwave-assisted nucleophilic substitution reaction of 3-amino-N-methylbenzenesulfonamide, 3-aminobenzenesulfonamide and N-(3-phenyl)methanesulfonamide with the appropriate key intermediate **4a–c** in ethanol (*Al-Sanea et al., 2019*).

## CDK8/CycC complex inhibition

The newly prepared phenylaminoquinolines **5a–i** were biologically evaluated as potential CDK8/CycC complex inhibitors. The percentage enzyme inhibition and half-maximal inhibitory concentration data of the target compounds with phenylaminoquinoline core structure and Staurosporine (as a standard inhibitor) against CDK8/CycC are summarized in Table 1.

## Molecular docking

For designing CDK8/CycC type I inhibitor, targeting the hinge residue is essential to inhibit the kinase activity of the complex. To visualize the binding interactions between the promising biologically active compound **5d**, we obtained a co-crystal structure of 6-isocyano-N-phenethylquinazolin-4-amine (Senexin A ) in complexation with CDK8:Cyclin C with the DMG motif in the "in" conformation at 2.2 Åresolution (PDB: 4F7S).

**Table 1** Inhibition data of CDK8/CycC for compounds 5a–i, using Staurosporine as a standard inhibitor.

| Comp. | R | R' | % enzyme inhibition ± SD (20 µM) | IC$_{50}$ (µM) |
|---|---|---|---|---|
| 5a | 6-Br | SO$_2$NH$_2$ | 83.91 ± 1.83 | 3.98 |
| 5b | 6-OCH$_3$ | SO$_2$NH$_2$ | 63.365 ± 2.31 | 5.34 |
| 5c | 7-Cl-6-F | SO$_2$NH$_2$ | 37.925 ± 1.31 | No inhibition |
| 5d | 6-Br | NHSO$_2$CH$_3$ | 88.59 ± 0.97 | 0.639 |
| 5e | 6-OCH$_3$ | NHSO$_2$CH$_3$ | 85.875 ± 0.94 | 1.42 |
| 5f | 7-Cl-6-F | NHSO$_2$CH$_3$ | 27.225 ± 8.46 | No inhibition |
| 5g | 6-Br | SO$_2$NHCH$_3$ | 22.425 ± 1.02 | No inhibition |
| 5h | 6-OCH$_3$ | SO$_2$NHCH$_3$ | 18.655 ± 0.48 | No inhibition |
| 5i | 7-Cl-6-F | SO$_2$NHCH$_3$ | 31.96 ± 0.82 | No inhibition |
| **Staurosporine** | | | | 1.00 E–4 |

## DISCUSSION

### Chemistry

Based on $^1$H-NMR, $^{13}$C-NMR spectroscopic data and high-resolution mass spectroscopy (HRMS), the structures of all newly synthesized compounds were confirmed. $^1$HNMR spectra of all finals **5a-i** showed new characteristic signals at $\delta$ 7.33–7.37 ppm, and 9.68–10.25 ppm corresponding to NH$_2$ and NH groups, respectively, that distinguished the finals **5a-i** from chloroquinolines **4a-c**. For compound **5d**, three characteristic signals at $\delta$ 2.42, 8.94, and 9.64 ppm were displayed and assigned to -NHCH$_3$, -SO$_2$NH- and -NH- protons, respectively.

### CDK8/CycC complex inhibition

According to the inhibition data stated in Table 1, the following structure–activity relationship (SAR) notes are described as follows.

The methanesulfonamide analogue **5d** showed maximal potency among all final compounds with submicromolar activity and IC$_{50}$ value of 0.639 µM, whereas, the corresponding primary benzenesulfonamide analogue **5a** exhibited a 6-fold decrease in potency (IC$_{50}$ = 3.98 µM). On the contrary, the corresponding substituted benzenesulfonamide analogue (**5g**) exhibited no CDK8/CycC complex inhibitory activity, confirming the crucial role of PKa values of sulfonamide groups for the intrinsic activity of pharmacophore of the phenylaminoquinoline scaffold–based compounds. Moreover, the primary benzenesulfonamide analogues (**5a & 5b**) exhibited single digit micromolar potency in inhibiting the CDK8/CycC. Whereas, methanesulphonamide analogues (**5d & 5e**) showed superior potency with IC$_{50}$ values of 0.639 and 1.42 µM, respectively. Noteworthy, all 7-chloro-6-fluoro substituted quinolines (**5c, 5f, 5i**) failed to inhibit the CDK8/CycC enzyme, signifying the remarkable adverse effects of some quinoline substituents on the binding interaction, and hence the intrinsic activity. Therefore, concerning the influence of substitution of the quinoline moiety, the inhibitory activities increased in the order of 7-Cl-6-F <6-OCH$_3$ <6-Br.
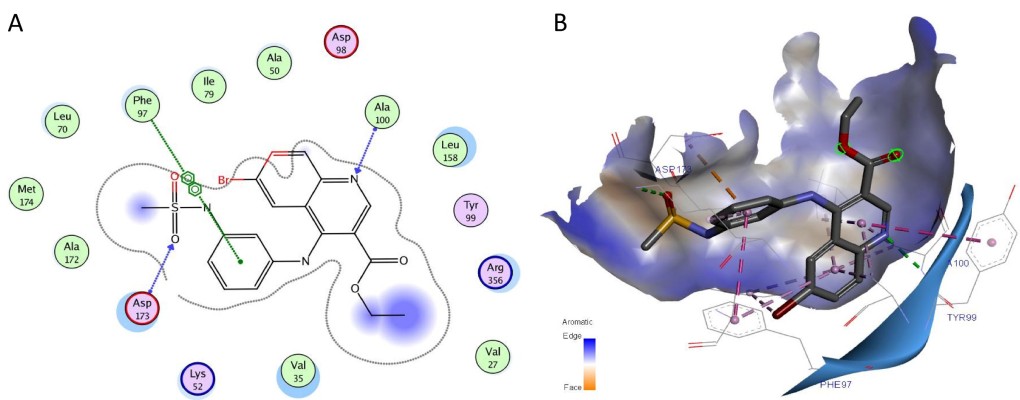

**Figure 1** (A) **Predicted 2D presentation of ligand binding modes of compound 5d in the kinase domain of CDK8/CycC active pocket. (B) Predicted 3D presentation of ligand binding pose of 5d in the active site of CDK8/CycC from the crystal structure 4F7S.** Discovery Studio Visualizer prepared 3D presentation by which the interacting residues are shown in lines; dotted lines are used to visualize the protein-ligand interactions.

## Molecular docking

The virtual docking study showed how the compound **5d** in the 3D docking pose can anchor in the kinase deep pocket and extended with diverse functional groups toward the hinge region and the front pocket. Two direct hydrogen bonds are formed between the inhibitor **5d** and the kinase domain of CDK8/CycC. The quinoline N forms an essential single H-bond with the backbone nitrogen of Ala100CDK8 on the hinge region. The sulfonyl O forms the second direct H-bond to the backbone N of Asp173CDK8 of the DMG motif. Moreover, $\pi-\pi$ stacking interaction with the gatekeeper residue (Phe97CDK8) and VDW interactions with several residues at the ATP binding pocket (Ala172CDK8, Ala50CDK8, Val27CDK8, Leu158CDK8, Val35CDK8, Tyr99CDK8, Ile79CDK8) were shown as depicted in Fig. 1.

## CONCLUSIONS

We have demonstrated a new series of phenylaminoquinoline core structure-based compounds. These target compounds **5a-i** have been synthesized and biologically evaluated as potential CDK8/CycC inhibitors. The methanesulfonamide analogues (**5d** & **5e**) proved to be the most potent compounds in suppressing CDK8/CycC enzyme, whereas, the un/substitutes benzenesulfonamide analogues showed inferior (or no) activity, demonstrating the advantage of highly acidic NH of sulfonamide moiety. Careful selection of quinoline moiety substituents is highly recommended, as 7-chloro-6-fluoro bearing analogues (**5c** & **5f**) showed no inhibition. Moreover, the secondary benzenesulfonamide analogues should be avoided, as they showed no inhibitory activities. We have discovered the most potent analogue **5d** with submicromolar potency against CDK8/CycC (IC$_{50}$ = 0.639 $\mu$M) and it can be prepared in four steps with an overall yield of 64% making it suitable for further investigations. Larger library of phenylaminoquinoline scaffold-based

analogues are going to be prepared by our team to establish detailed and distinguished SARs.

In summary, the novel ((3-(N-methylsulfamoyl)phenyl)amino)quinoline derivatives exhibited potent CDK8/CycC inhibitory activity. Developing inhibitors targeting CDK8/CycC offers an exciting approach for treatment of human carcinomas.

## ACKNOWLEDGEMENTS

I would like to express my special appreciation and thanks to my colleague Dr. Ahmed Elkamhawy, you have been a great research collaborator for me in this project.

### Funding

This research was supported by Deanship of Scientific Research at Jouf University (Project no. 144/40) and by the Korea Institute of Science and Technology (2019 KIST School Partnership Research Grant). The funders had no role in study design, data collection and analysis, decision to publish, or preparation of the manuscript.

### Grant Disclosures

The following grant information was disclosed by the author:
Deanship of Scientific Research at Jouf University: 144/40.
Korea Institute of Science and Technology.
2019 KIST School Partnership Research Grant.

### Competing Interests

The author declares there are no competing interests.

### Author Contributions

- Mohammad M. Al-Sanea conceived and designed the experiments, performed the experiments, analyzed the data, prepared figures and/or tables, authored or reviewed drafts of the paper, and approved the final draft.

### Data Availability

Data is available in the Supplemental Files.

### Supplemental Information

Supplemental information for this article can be found online at http://dx.doi.org/10.7717/peerj.8649#supplemental-information.

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
