# Peer review of "Synthesis and biological evaluation of small molecule modulators of CDK8/Cyclin C complex with phenylaminoquinoline scaffold"

_PeerJ, doi:10.7717/peerj.8649_

## Round 0.1 · original submission · Minor Revisions

Please attend the reviewer's comments, in a detailed rebuttal letter and a tracked-changed document. It would be important to provide further explanation of the docking experiments, implicate if contact residues are known to have mutations, resistance to staurosporine or other important biological significance of the results.

Reviewer 1 ·

Basic reporting

The writing is clear and well structured. Contains sufficient revised bibliography regarding the subject of the document. The author has several reports in the area. The support material presents the characterization of the compounds reported in the document.

Experimental design

Due to the structure of the article, first presents the synthesis of the compounds, but reference to scheme 1 until results, so it is difficult to follow the document. I recommend that scheme 1 be referenced in the synthesis.
In line 103 it is recommended to indicate in which articles the compounds 3-4 and 5a-c, i were previously reported.

The way of reporting the characterization of the compounds is not homogenous. It is recommended to follow a single style and completely assign the protons. One suggestion is to enumerate the protons and indicate it. After each value of J, it indicates that they are in Hz. Some have the error of being in this way HZ and must be Hz.

Figure 1, requires a higher resolution.
In Table 1, it is recommended that significant numbers be considered, for example
The values for 5a es correct (83.91 ± 1.83), but the values for 5b its not correct (37.925 ± 1.31), the way of reporting must be homogeneous.

Validity of the findings

The results presented are interesting and of great value for the study area. The conclusions are in accordance with what is presented in the document.

Additional comments

I believe that the document is well structured and of great interest to the area. I believe that it would be improved if it reports in a homogeneous way, since it reports synthesis of new compounds and must completely assign the signals of each compound. It would be interesting to discuss more the structure-response relationship.

Reviewer 2 ·

Basic reporting

CDK8/CycC complex has been recognized as a potential target for cancer therapy. This manuscript reported the synthesis and evaluation of a series inhibitor of CDK8/CycC complex. Generally, the manuscript is clear and well written with professional English. However, the literature references are outdated. The figure legend doesn't provide sufficient details.

Experimental design

The experiments are original and well performed.

Validity of the findings

The results can clearly stated, conclusions are well stated which could be supported by their results.

Additional comments

1. The authors should be careful about some typo mistakes in the manuscript. For example, in the first paragraph of introduction.
2. Figure legend is too simple. The authors should include more information.
3. The references are outdated, they should cite more lately references.
4. Include more details in the results part.

---

## Round 0.2 · Minor Revisions

There are several typos and format issues to work out. For example, references are included outside of the ending point of a sentence (see lines 58-59, also lines 64-65) and several typos. Typo on line 94.
Also, please homogenize use superscripts, in some places is used 1H or 13C and later with superscripts (line 146). I consider superscripts that should be used.

The sentence on line 366 "Kinase Profiling is 10 dose IC50 singlet assay" is not understandable.

In conclusions and in the abstract, a general public explanation of the main findings is required. PeerJ has a wide audience and the conclusions are addressed to a medicinal chemistry audience.

Since accepted manuscripts still have a final check on language and style, it may be useful to get it checked by a professional proofreading service to handle those details. If that is the case, please send a copy of the certificate of revision.

---

## Round 0.3 · accepted · Accept

The manuscript has improved over the review rounds and it is now accepted at PeerJ. In case that the PeerJ staff finds some technical issues on the manuscript, you will be contacted again.

Congratulations,
Rogerio Sotelo-Mundo